# Impact of Enhanced Recovery after Surgery with Preoperative Whey Protein-Infused Carbohydrate Loading and Postoperative Early Oral Feeding among Surgical Gynecologic Cancer Patients: An Open-Labelled Randomized Controlled Trial

**DOI:** 10.3390/nu12010264

**Published:** 2020-01-20

**Authors:** Ho Chiou Yi, Zuriati Ibrahim, Zalina Abu Zaid, Zulfitri ‘Azuan Mat Daud, Nor Baizura Md. Yusop, Jamil Omar, Mohd Norazam Mohd Abas, Zuwariah Abdul Rahman, Norshariza Jamhuri

**Affiliations:** 1Department of Nutrition and Dietetics, Faculty of Medicine and Health Sciences, Universiti Putra Malaysia, Seri Kembangan 43400, Malaysia; 2Department of Dietetics and Food Service, National Cancer Institute, Ministry of Health, 4, Jalan P7, Presint 7, Putrajaya 62250, Malaysia; 3Department of Surgical Oncology, National Cancer Institute, Ministry of Health, 4, Jalan P7, Presint 7, Putrajaya 62250, Malaysia

**Keywords:** enhanced recovery after surgery, whey protein carbohydrate loading, gynecologic cancer

## Abstract

Enhanced Recovery after Surgery (ERAS) with sole carbohydrate (CHO) loading and postoperative early oral feeding (POEOF) **s**hortened the length of postoperative (PO) hospital stays (LPOHS) without increasing complications. This study aimed to examine the impact of ERAS with preoperative whey protein-infused CHO loading and POEOF among surgical gynecologic cancer (GC) patients. There were 62 subjects in the intervention group (CHO-P), which received preoperative whey protein-infused CHO loading and POEOF; and 56 subjects formed the control group (CO), which was given usual care. The mean age was 49.5 ± 12.2 years (CHO-P) and 51.2 ± 11.9 years (CO). The trial found significant positive results which included shorter LPOHS (78.13 ± 33.05 vs. 99.49 ± 22.54 h); a lower readmission rate within one month PO (6% vs. 16%); lower weight loss (−0.3 ± 2.3 kg vs. −2.1 ± 2.3 kg); a lower C-reactive protein–albumin ratio (0.3 ± 1.2 vs. 1.1 ± 2.6); preserved muscle mass (0.4 ± 1.7 kg vs. −0.7 ± 2.6 kg); and better handgrip strength (0.6 ± 4.3 kg vs. −1.9 ± 4.7 kg) among CHO-P as compared with CO. However, there was no significant difference in mid-upper arm circumference and serum albumin level upon discharge. ERAS with preoperative whey protein-infused CHO loading and POEOF assured better PO outcomes.

## 1. Introduction

The primary modalities of cancer treatment are surgery, chemotherapy, and radiotherapy; these may be used alone or in combination. For those cancers which are well margined and operable, surgery is the first treatment for gynecologic cancer (GC) [1]. The conventional feeding strategy for pre- and post-operation is for prolonged fasting or rest for both the patient and the gastrointestinal tract, which will enhance the organic response to surgical trauma and subsequently, delay the postoperative recovery [2]. Inadequate oral intake due to delayed oral feeding causes depletion of nutrient storage in a patient’s body [3]. This is because of the utilization of energy which is converted from protein sources of the body (muscle) and promotes catabolism. Thus, patients experience weight loss and muscle mass loss postoperatively [4].

After realizing that a unimodal intervention was impractical to solve multimodal perioperative morbidity issues as in the conventional surgery approach, Kehlet and Wilmore initiated and developed the concept of a multimodal perioperative protocol, named Fast Track Recovery surgery [5]. This concept later led to the development of ERAS (Enhanced Recovery after Surgery) into more comprehensive protocol by Ljungqvist and colleagues [6]. The evidence-based ERAS program was designed to reduce perioperative stress, maintain postoperative physical function, and accelerate postoperative recovery [5]. By using a multimodal stress-minimizing approach, repeated convincing postoperative outcomes including a reduction in the morbidity rate, improved recovery, and shortened length of hospital stay (LOS) were found after major colorectal surgery [5]. There was a strong consensus that recommended shortening preoperative fasting via carbohydrate loading and postoperative early oral feeding for patients who undergo a major operation with the aim of achieving better postoperative outcomes [7].

As per the recommendation in ERAS and European Society for Parenteral and Enteral Nutrition (ESPEN) 2017, shortening the preoperative fasting via carbohydrate (CHO) loading and postoperative early initiation of oral feeding is preferable, with improved postoperative outcomes and without increased postoperative readmission rate [1,5]. To date, carbohydrate loading with solely carbohydrate drink was widely used and has been shown to have beneficial outcomes [1,6]. ERAS has increasingly been practiced in colorectal surgeries [8], gastrointestinal surgery [9], and various other fields [10,11]. Studies highlighted that the ERAS includes providing patients with just CHO loading preoperatively and allows early oral feeding with clear fluid, followed by solid food postoperatively [5,7,12,13]. For aspects of GC surgery, the ERAS protocol with preoperative CHO loading and postoperative early oral feeding also showed a significant reduction in length of stay (LOS) without increasing readmission and complication rates [10,12,14,15]. In Malaysia, surgeries which implemented the ERAS protocol with preoperative CHO loading and postoperative early oral feeding showed positive postoperative outcomes, including in colorectal resection [14], pancreaticoduodenectomy [16], and liver resection [15]. However, the impact of the ERAS protocol in surgical GC remains unclear in Malaysia.

Other than calories and CHO, protein is crucial for postoperative recovery as it promotes anabolism, slows down muscle catabolism, and shortens the inflammatory phase [1]. Whey protein can be absorbed and utilized by skeletal muscle during stress and stimulates protein synthesis because it contains essential amino acids, especially branch-chain amino acids which are characterized by a high degree of digestibility and rapid absorption in the small bowel [17]. To explore the role of protein in CHO loading under the ERAS protocol, studies in cholecystectomy [10], gastrointestinal cancer [9], and oral and maxillofacial surgery [11] resulted in a shortened postoperative length of stay and reduced postoperative inflammatory reaction. Perrone and da Silva Filho demonstrated that loading in the form of whey protein infused with CHO drinks prior to operations on cholecystectomy patients preserved postoperative muscle strength, improved patients’ satisfaction by reducing fatigue, anxiety and discomfort, and reduced the trauma-induced inflammatory response [9].

Guidelines for preoperative CHO loading and postoperative early oral feeding in gynecologic cancer patients under ERAS setting were updated [18]. However, the impact of providing whey protein-infused CHO loading drinks as preoperative CHO loading among surgical GC patients remains unclear. Thus, there was a need to fill the research gap regarding the effect of the ERAS protocol with preoperative whey protein-infused CHO loading and postoperative early oral feeding in surgical GC patients.

Moreover, most ERAS protocol studies were focused on postoperative clinical outcomes such as LOS, complication, and the readmission rate [4,5,15,16,19,20]. However, the role of the ERAS protocol in the changes of nutritional aspect (body composition) and functional status (handgrip strength) remain unclear. This study aims to determine the impact of ERAS with a preoperative whey protein-infused CHO-loading drink and postoperative early oral feeding on postoperative outcomes and complications, as well as the nutritional status and function status among surgical GC patients.

## 2. Materials and Methods

This study was an open-labelled randomized controlled trial (RCT) conducted at the National Cancer Institute, Putrajaya, Malaysia. The study was primarily conducted in the Surgical Gynecology Department, which is a multidisciplinary clinic and female surgical ward. The RCT conformed to the Consolidated Standards of Reporting Trials (CONSORT) statement for reporting RCT with two arms comparing an intervention group with a control group [21]. ERAS protocol was initiated in the current setting with this study (as intervention group). After consenting to participate, subjects were allocated into intervention (CHO-P) and control (CO) groups randomly. The study protocol was approved by the Malaysia Research Ethic Committee (MREC) with ethic approval number NMRR-17-1070-36021 (IIR) and was registered on ClinicalTrials.gov with identity number NCT03667755.

### 2.1. Sample Size Calculation

The study used a power analysis to determine the sample size needed to detect a difference between the intervention and control groups in the primary outcomes (postoperative outcomes). The sample size of the study was calculated by using the formula proposed by Woodword [22]. According to the results of a previous study by Balayla et al. [4], the calculated sample size was 33 subjects per group. After adjustment for an 80% response rate and 90% expected eligibility rate (power = 80%, alpha level = 0.05), and to account for a 20% drop-out rate, a total of 110 subjects (55 subjects in each group) was needed.

### 2.2. Subjects

A total of 118 subjects fulfilled the eligibility criteria and enrolled in the study (Figure 1). Ambulated Malaysian patients aged over 18 years and scheduled for elective surgery for suspected GC were included in this study. Those who were allergic to soy or whey protein; diagnosed with chronic kidney diseases, ischemic heart disease or diabetes mellitus; or involved in other intervention studies were excluded. Oral and written information about the study procedures were provided by a gynecologic surgeon or dietitian in an undisturbed consultation room. The patient was recommended to take at least 24 h to consider and discuss their participation with family members before deciding to enlist. A Patient Information and Study Consent Form was given to eligible patients in order to ease the discussion with family members. Written informed consent forms were collected on admission day for recruitment into the dietetic clinic. 

### 2.3. Study Design and Randomization

This was an open-labelled randomized control trial. Hence, a blind step was not included in this trial. Consenting subjects were randomized into two groups before baseline assessment: intervention (CHO-P) and control group (CO). Randomization was done by a computer-generated number randomization which was prepared by an independent statistician. The allocation of randomized numbers was concealed in sealed envelopes by the study coordinator. The envelope was only opened after consent and before the baseline assessment. The comparison of study protocol between the intervention group (CHO-P) under Enhanced Recovery after Surgery and control group (CO) under conventional perioperative care is shown in Table 1. The RECOvER checklist of items was used for reporting of ERAS compliance, outcomes and element research (Appendix A) [23].

### 2.4. Intervention Group (CHO-P)

Measurement of the anthropometric and dietary assessment was made on the day of admission in the dietitian clinic. After admission, subjects were under the multimodal ERAS perioperative protocol. Subjects were provided with a standardized specially formulated drink in the evening 12 h before the operation and 3 h before operation. The drink that was provided in the evening before the operation consisted of 500 kcal, 100 g carbohydrate, and 18 g whey protein (total 474 mL), whilst the drink provided 3 h before the operation comprised 237 mL and provided 250 kcal, 50 g carbohydrate, and 9 g whey protein in a lactose-free, clear, tea-colored, fruit-flavored fluid. After the operation, subjects were instructed to omit solid food for 6 h. However, subjects were allowed to consume specially formulated clear fluid (474 mL, providing 500 kcal, 100 g carbohydrate, and 18 g whey protein) 4 h post-operation (without the presence of bowel sounds). When they could tolerate at least 500 mL of specially formulated clear fluid with another additional clear fluid, solid food was permitted. The staff nurse in charge monitored anesthetic risks of drinking whey protein and ensured compliance of the subjects with the drink provided.

### 2.5. Control Group (CO)

Subjects were required to attend the dietitian clinic to have anthropometry and dietary assessments on the same day of admission. After admission, subjects were under conventional perioperative care. Their last meal was dinner, which was a minimum of 12 h before operation. Subjects started fasting from midnight on the day of operation. On the post-operation day, subjects were reviewed by a gynecologic surgeon. They were allowed clear fluid once there were bowel sounds. After subjects tolerated clear fluid, they proceeded with nourishing fluid, then soft diet, and they only received a regular solid diet after tolerating the soft diet.

### 2.6. Discharge Criteria

The gynecologic surgeon determined time to discharge of subjects based on an assessment of discharge criteria such as optimal pain management (orally), the ability to ambulate independently, adequate nutrition intake, gastrointestinal function return, and without suspicion of complications [24].

### 2.7. Data Collection Procedure

Data were collected at baseline (upon admission), during postoperative hospitalization, upon discharge post-operation, and at one-month post discharge.

### 2.8. Outcomes Measurement

The primary outcomes were the postoperative outcomes (length of postoperative hospital stay, clear fluid toleration, food toleration, and bowel function return) between the CHO-P and CO. The length of postoperative hospital stay was defined as the time from the operation end to discharge from the hospital. The length of clear fluid toleration was defined as the time from operation end to when the patient could tolerate clear fluid. The length of food toleration was defined as the time from operation end to tolerance of regular food. Length of bowel function return was defined as the time from operation end to flatus or bowel opening. Primary outcomes were assessed by a gynecologic surgeon and recorded on a data collection form (progress in ward form) by the nurse in charge.

Secondary outcomes were postoperative complications including postoperative nausea and vomiting (PONV), ileus, and infection; these were monitored and recorded by the surgical gynecologic team. Overall complications were assessed during hospital stay; complication rates were defined per patient, as discussed elsewhere [25]. Readmissions of study subjects were documented from the day of discharge until 1-month post-operation (30 days). Readmission complications were described separately from the complications during hospital stay.

Other outcomes measures included the changes of body composition (weight, muscle mass, fat mass, fat-free mass, and mid-upper arm circumference) via body composition analyzer TANITA model SC 300 and flexible, nonstretchable SECA measuring tape model 201, nutritional assessment (the Scored Patient-Generated Subjective Global Assessment (PG-SGA)), biochemical profile (hemoglobin, C-reactive protein, albumin, and C-reactive protein-albumin ratio (CAR)), and functional status (handgrip strength) via JAMMAR dynamometer.

### 2.9. Adverse Events and Data Safety Monitoring

There were no serious side effects identified and low anesthetic risk of drinking whey protein three h before operation (vomiting/nausea) in this study. The staff nurse in charge monitored subjects if there were any adverse events after consuming the study product. If there were any adverse events/intercurrent illnesses, the staff nurse in charge reported to the medical officer in charge (gynecology) immediately. The medical officer initiated the appropriate treatment. Study findings potentially improved treatment outcomes. The expected benefit outweighed the minimal risk to subjects and, thus, this study should be supported. There was no prorated payment for participation in this study.

### 2.10. Statistical Analysis

The analyses were performed using IBM SPSS (version 23.0). All randomized RCT subjects were included in analysis on an intention-to-treat (ITT) basis. According to Gupta [26], ITT analysis reflects the practical clinical scenario because it admits noncompliance and protocol deviations, maintains prognostic balance generated from the original random treatment allocation, and gives an unbiased estimate of treatment effect. The results are presented as mean ± standard deviation, while categorical data are presented as frequencies and percentages. Comparisons of numerical data which were normally distributed between two groups were analyzed using the independent t-test. Pearson’s Chi-square test was used to test differences among categorical data. All probability values used were two-sided with *p* < 0.05 considered statistically significant.

## 3. Results

The compliance study protocol between the intervention group (CHO-P) under Enhanced Recovery after Surgery and control group (CO) under conventional perioperative care is shown in Table 2. The CHO-P complied to the ERAS protocol, while the CO was under conventional care. The subjects’ demographic and baseline characteristics are shown in Table 3 and Table 4. The majority of subjects were Malay (78.8%) and diagnosed with ovarian cancer (42.3%) at stage one (87.2%). The mean age was 49.5 ± 12.2 years for CHO-P and 51.2 ± 11.9 years for CO. The PG-SGA score was 6.7 ± 5.2 in CHO-P and 7.0 ± 5.5 in CO. CHO-P and CO groups experienced −4.5% ± 6.8% and −5.3% ± 7.2% weight loss past one month, respectively. There was no statistically significant difference among any baseline variables between groups. CHO-P achieved better postoperative outcomes when compared with CO (Table 4). CHO-P recorded significantly shorter length of postoperative stay (LPOHS, 78.13 ± 33.05 h) when compared with CO (99.49 ± 22.54 h), *p* < 0.01; the length of clear fluid toleration in the CHO-P group (10.23 ± 3.42 h) was shorter than that of CO (21.89 ± 8.77 h), *p* < 0.01. The CHO-P group also tolerated solid diet postoperatively in a shorter time (22.05 ± 11.70 h) compared with CO (52.90 ± 16.43 h), *p* < 0.01. Moreover, the CHO-P group reported significantly (*p* < 0.01) faster bowel function return (28.32 ± 19.06 h) and bowel opening (36.04 ± 21.71 h) than CO (53.10 ± 17.29 h and 68.84 ± 19.09, respectively) (Table 5). The CO group documented significantly (*p* < 0.01) higher postoperative complications, which included more postoperative nausea and vomiting, and also had an increased readmission rate within 30 days post-operation (16%) compared with CHO-P (6%), *p* < 0.05 (Table 6). Reasons for readmission were infection (54%) and wound debridement (46%). The CHO-P group preserved nutritional status significantly better (*p* < 0.01), had a significantly better biochemical profile (acute phase inflammatory marker) (*p* < 0.05), and secured functional status significantly faster (*p* < 0.04) compared with the CO group (Table 7).

## 4. Discussion

The findings of this study showed that the multimodal ERAS approach with preoperative whey protein-infused carbohydrate loading and postoperative early oral feeding resulted in positive postoperative outcomes, less nutritional depletion, greater muscle mass conservation and strength, and concealed inflammatory response. Positive outcomes also included a shortened length of postoperative hospital stay without increasing complications among surgical GC patients. Moreover, this approach assures a better postoperative body composition, muscle strength preservation, and suppresses acute-phase inflammatory markers postoperatively.

This ERAS protocol study demonstrated significant reductions in LPOHS for surgical GC patients. The reduction of 21.3 h (0.88 days) in LPOHS for surgical GC patients represents not only a statistically significant result but also a clinically meaningful one. This was in line with some other studies that reported that the ERAS protocol with preoperative whey protein-infused CHO loading and postoperative early oral feeding approach shortened LPOHS [6,8,9]. Plausible explanations include that the ERAS approach results in earlier food toleration, mobilization, and bowel function return [27]. Improved postoperative outcomes could be obtained from CHO loading and early oral feeding by a reduction in surgical-induced altered metabolism and by providing early nutrition supply postoperatively [13]. Andersen et al. emphasized that keeping patients nil-by-mouth postoperatively did not bring beneficial effects [28]. Early oral feeding postoperatively did not cause any detrimental effects but reduced the incidence of wound infection, pneumonia, intra-abdominal abscess, and mortality [19].

Other ERAS protocol studies in GC which used only postoperative early oral feeding, and not CHO loading, resulted in reduced abdominal distension, nausea, and vomiting, and also hastened bowel recovery [29,30]. A study has demonstrated that prolonged fasting diminished the gastrointestinal peristalsis and caused irregular contraction waves instead of frequent eating peristalsis and regular contraction waves [31]. Therefore, postoperative early oral feeding not only accelerates intestinal function recovery, but also provides nutrients needed during recovery to maintain the intestinal mucosal barrier function and further accelerate organ recovery [8,32].

The one-month post-discharge readmission rate for the intervention group was significantly lower than that of the control group, which was consistent with a study by Marx et al., which demonstrated that the ERAS protocol in ovarian cancer surgery did not increase the readmission rate and had no acute life-threatening readmissions in the intervention group [33]. Importantly, shortened postoperative hospital stays did not come at the price of increased hospital readmissions [33].

This study did not find any aspiration pneumonia or increased gastric volume, so our findings support the feasibility of implementation of preoperative whey protein-infused CHO loading 3 h prior to operation. There were multiple RCTs similar with the present study that recommended reducing the period of preoperative fasting to 2 h with clear fluid or CHO-enriched drinks [5,10,19,27]. Brady et al. stated that absorption of a small liquid meal was efficient in less than 90 min [27]. Other than solely CHO-loading drink, there were studies which investigated the use of CHO with protein as the form of CHO-loading on postoperative outcomes. Riley et al. proved that ingestion of a clear liquid which contained a protein and CHO supplement at least 2 h before gynecologic surgery did not increase gastric volume or decrease gastric pH [34]. Moreover, this finding was consistent with previous reports by Henriksen et al., Perrone et al., and Watanabe et al., who found that a drink containing CHO and protein 2–3 h pre-operation was feasible without increasing aspiration risk when compared with the prolonged conventional preoperative fasting period [6,9,35].

In this study, the ERAS protocol with preoperative whey protein-infused CHO loading and postoperative early oral feeding focused on the nutritional outcomes as well. There was a better nutritional outcome, including minimal reduction in body weight and fat mass, as well as preservation of muscle mass and handgrip strength, throughout hospitalization. Adequate protein intake plays an important role during postoperative recovery by promoting anabolism, minimizing protein deficit and muscle catabolism, and reducing the inflammatory phase [17]. Whey protein, one protein type, is categorized as a fast-digesting protein as it can be absorbed and utilized easier by skeletal muscle during stress and stimulates protein synthesis because it contains essential amino acids, especially branch-chain amino acids, which are characterized by a high degree of digestibility and rapid absorption in the small bowel [36]. Thus, adding whey protein in preoperative CHO loading could reduce further the postoperative insulin resistance that otherwise leads to an increase in catabolism of skeletal muscle and results in body fat and protein loss [9]. Early oral feeding postoperatively provided early energy protein supply and subsequently reduced the risk of further catabolism due to a negative energy protein balance [32].

This study found that the inflammatory response after surgery occurred in both groups, but the CHO-P group showed a less intense response. Preoperative whey protein-infused CHO loading and postoperative early oral feeding did reduce the inflammatory response [9]. The current study also demonstrated that a more intense inflammatory response might contribute to prolonged LPOHS, which is similar to findings of a study by Pexe-Machado et al. [8]. This study found that the postoperative muscle strength was higher in the CHO-P group than in the CO group, which is consistent with findings of a study by Noblett et al., which found better muscle strength preservation in the preoperative CHO-loading group when compared with a fasting group among colorectal cancer patients [37]. Henriksen et al. also concluded that preoperative CHO plus protein loading acquired higher muscle strength if compared with control group [6]. The preservation of muscle strength was due to the attenuated insulin resistance and subsequently reduced protein breakdown [38].

The strength of the current study is that it is the first study to evidence a multimodal ERAS approach with preoperative whey protein-infused CHO loading and postoperative early oral feeding, which was shown to result in better nutritional outcomes, a suppressed acute-phase inflammatory response, and the preservation of muscle strength among surgical GC patients in Malaysia. Nevertheless, the study had several limitations: The postoperative observation was limited to one month, so any long-term effects of the ERAS protocol with preoperative whey protein-infused CHO loading and postoperative early oral feeding on wound healing were not assessed. Furthermore, this was a single-center study, and the protocol used might not be applicable to other hospitals.

## 5. Conclusions

As conclusion, high compliance on the multimodal ERAS with preoperative whey protein-infused CHO loading and postoperative early oral feeding shortened the length of postoperative hospital stay without increasing complications among surgical GC patients. Therefore, whey protein-infused CHO drink as carbohydrate loading, rather than a solely CHO drink, and postoperative early oral feeding are suggested to be implemented into multidisciplinary ERAS protocol in order to further assure a better postoperative nutritional outcome, muscle strength preservation, and suppressed acute-phase inflammatory markers postoperatively. To confirm our study findings, a long-term study which extends the observation (nutrition status and functional status) period to one month and three months post-operation in a multicenter setting is suggested.

## Figures and Tables

**Figure 1 nutrients-12-00264-f001:**
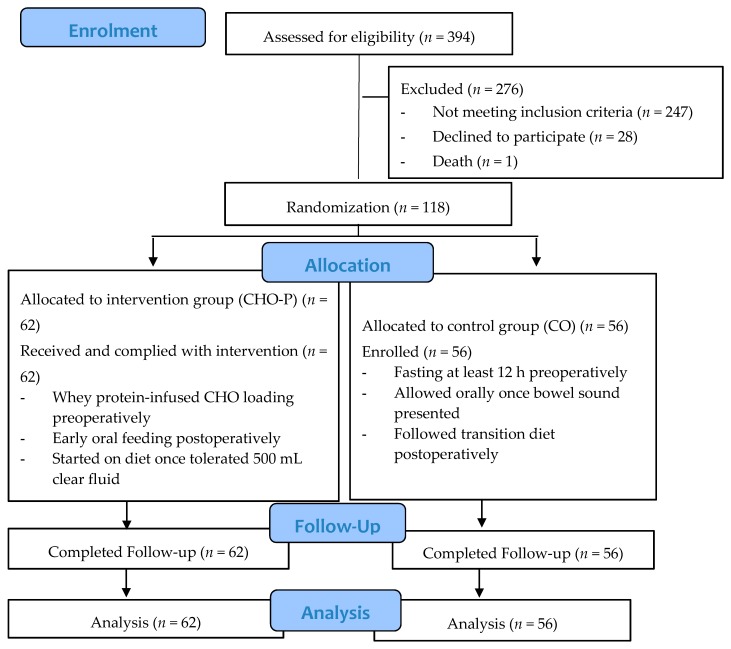
Flow diagram for subject screening and recruitment.

**Table 1 nutrients-12-00264-t001:** Comparison of the study protocol between the intervention group (CHO-P) under Enhanced Recovery after Surgery* and control group (CO) under conventional perioperative care.

Items	CHO-P	CO
(a) preadmission patient education regarding the protocol	Subjects were counselled and had the ERAS protocol explained by a surgeon, anesthetist, and dietitian	Preoperative counselling by surgeon and anesthetist
(b) preadmission screening and optimization as indicated for nutritional deficiency, anemia, HbA1c, tobacco cessation, and ethanol use	Preadmission screening: nutritional deficiency via scored PG-SGA, anemia, tobacco cessation, and ethanol counselling	Preadmission screening: anemia, tobacco cessation, and ethanol counselling
(c) fasting and carbohydrate loading	Normal diet until 6 h before operation; in the evening before operation the drink comprised 500 kcal, 100 g carbohydrate, and 18 g whey protein (total 474 mL), whilst the drink provided 3 h before operation comprised 237 mL with 250 kcal, 50 g carbohydrate, and 9 g whey protein	Last meal was dinner, which was minimal 12 h before operation. Subjects started fasting from midnight on the day of operation
(d) Pre-emptive analgesia (dose, route, timing)	No routine	
(e) Antiemetic prophylaxis (dose, route, timing)	8 mg Dexamethasone given intravenously prior to operation	
(f) Intraoperative fluid management strategy	Fluid maintenance given according to total body fluid loss	
(g) types, doses, and routes of anesthetics administrated	Continuous intravenous Rocuronium	
(h) patient warming strategy	Warm blanket and intravenous warmer	
(i) Management of postoperative fluids	0.5 mL/kg/h for 6 h	
(j) Postoperative analgesic and antiemetic plans	Paracetamol 1 g every 6 h orally, Maxalon 10 mg every 8 h intravenously	
(k) plan for opioid minimization	Multimodal oral medicine: paracetamol 1 g every 6 h and Celebrex 200 mg every 12 h	
(l) drain and line management	No routine—wound drains	
(m) Early mobilization strategy	Enforced to stay out of the bed by postoperative day 1	
(n) postoperative diet and bowel regimen management	Subjects were allowed to consume specially formulated clear fluid (474 mL, 500 kcal, 100 g carbohydrate, and 18 g whey protein) 4 h post-operation (without presence of bowel sound). When they could tolerate at least 500 mL of specially formulated clear fluids with additional other clear fluid, solid diet was permitted.	Subjects were allowed clear fluid once there were bowel sounds. After tolerating clear fluid, they proceeded with nourishing fluid, then soft diet, and they only received a regular solid diet after tolerating the soft diet.
(o) criteria for discharge	Optimal pain management orally, able to ambulate independently, adequate nutrition intake, gastrointestinal function return, and without suspicion of complications	
(p) tracking of post-discharge outcomes	Follow up in multidisciplinary clinic within 7 days post-discharge. If any emergency before follow up, subjects attended emergency department NCI.	

*RECOvER checklist [23]. ERAS: Enhanced Recovery after Surgery. PG-SGA: scored patient-generated subjective global assessment. HbA1c: Hemoglobin A1c.

**Table 2 nutrients-12-00264-t002:** Compliance study protocol between the intervention group (CHO-P) under Enhanced Recovery After Surgery and control group (CO) under conventional perioperative care.

Items	CHO-P	CO
(a) preadmission patient education regarding the protocol	100%	0%
(b) preadmission screening and optimization as indicated by nutritional deficiency, anemia, HbA1c, tobacco cessation, and ethanol use	100%	50%
(c) fasting and carbohydrate loading	100%	0%
(d) Pre-emptive analgesia (dose, route, timing)	Not applicable	Not applicable
(e) Antiemetic prophylaxis (dose, route, timing)	100%	100%
(f) Intraoperative fluid management strategy	100%	100%
(g) types, doses and routes of anesthetics administrated	100%	100%
(h) patient warming strategy	100%	100%
(i) Management of postoperative fluids	100%	100%
(j) Postoperative analgesic and antiemetic plans	100%	100%
(k) plan for opioid minimization	100%	100%
(l) drain and line management	100%	100%
(m) Early mobilization strategy	96.7%	76.8%
(n) postoperative diet and bowel regimen management	100%	0%
(o) criteria for discharge	100%	100%
(p) tracking of post-discharge outcomes	100%	100%

**Table 3 nutrients-12-00264-t003:** Demographic characteristics of intervention group (CHO-P) and control group (CO).

Characteristics	CHO-P (*n* = 62)	CO (*n* = 56)
	*n* (%)	*n* (%)
**Ethnic**		
Malay	46 (55)	47 (45)
Chinese	10 (45)	12 (55)
Indian	6 (46)	7 (54)
**Diagnosis**		
Ovarian cancer	24 (48)	26 (52)
Endometrial cancer	22 (55)	18 (45)
Cervical cancer	13 (62)	8 (38)
Uterine cancer	3 (43)	4 (57)
**Stage**		
1	55 (53)	48 (47)
2	3 (50)	3 (50)
3	1 (33)	2 (67)
Advanced	3 (50)	3 (50)
**Family history**		
Yes	21 (48)	23 (52)
No	41 (55)	33 (45)
**ASA score**		
1 (Normal healthy)	28 (61)	18 (39)
2 (Mild systemic disease)	33 (48)	36 (52)
3 (Severe systemic disease)	1 (33)	2 (67)
**Procedure**		
TAHBSO	37 (48)	40 (52)
Salpingoophrectomy	16 (67)	8 (33)
Radical Hysterectomy	6 (55)	5 (45)
Debulking Tumor	3 (50)	3 (50)

Abbreviations: ASA: American Society of Anesthesiologists score, TAHBSO: total abdominal hysterectomy bilateral salpingooperectomy.

**Table 4 nutrients-12-00264-t004:** Baseline clinical characteristics of intervention group (CHO-P) and control group (CO).

Characteristics	CHO-P (*n* = 62) Mean ± SD	CO (*n* = 56) Mean ± SD	*p-*Value
Age	49.5 ± 12.2	51.2± 11.9	0.447
**Biochemical Profile**			
Serum albumin (g/L)	39.1 ± 5.3	37.3 ± 6.3	0.097
CRP (mg/L)	14.2 ± 26.9	24.2 ± 57.5	0.238
CAR	0.4 ± 1.0	0.9 ± 2.6	0.169
Hemoglobin (g/L)	11.9 ± 1.5	11.6 ± 1.9	0.061
**Body composition**			
Weight (kg)	63.8 ± 13.3	66.4 ± 16.7	0.361
BMI (kg/m^2^)	26.0 ± 6.0	27.2 ± 6.4	0.310
Muscle mass (kg)	37.2 ± 4.1	37.4 ± 4.7	0.765
Fat mass (kg)	24.4 ± 9.4	26.2 ± 12.9	0.383
Fat Free Mass (kg)	39.5 ± 7.5	40.1 ± 5.1	0.499
MUAC (cm)	28.0 ± 4.1	28.8 ± 6.0	0.403
Weight changes in past 1 month (kg)	−3.1 ± 4.4	−3.7 ± 5.1	0.474
Percentage weight changes in past 1 month (%)	−4.5 ± 6.8	−5.3 ± 7.2	0.472
**Nutritional assessment**			
PG-SGA score	6.7 ± 5.2	7.0 ± 5.5	0.758
**Functional status**			
Handgrip strength (kg)	16.7 ± 6.1	15.0 ± 6.2	0.123

Independent *t*-test; *p*-value < 0.05. Abbreviations: CRP: C-reactive protein, CAR: C-reactive protein−albumin ratio; MUAC: mid-upper arm circumference; PG-SGA: patient-generated subjective global assessment. BMI: body mass index.

**Table 5 nutrients-12-00264-t005:** Comparison of postoperative outcomes of surgical gynecologic cancer (GC) between intervention (CHO-P) and control groups (CO).

Post-Operative Outcomes	CHO-P (*n* = 62) Mean ± SD	CO (*n* = 56) Mean ± SD	*t*-Value	*p*-Value
LPOHS (h)	78.13 ± 33.05	99.49 ± 22.54	−4.056	0.000 **
LOCF (h)	10.23 ± 3.42	21.89 ± 8.77	−9.329	0.000 **
LOSDT (h)	22.05 ± 11.70	52.90 ± 16.43	−11.633	0.000 **
LOBFR (h)	28.32 ± 19.06	53.10 ± 17.29	−7.368	0.000 **
LOBO (h)	36.04 ± 21.71	68.84 ± 19.09	−8.678	0.000 **

Independent *t*-test. ** *p* < 0.001; Abbreviations: LPOHS: length of postoperative stay; LOCF: length of clear fluid toleration; LOSDT: length of solid diet toleration; LOBFR: length of bowel function return (flatus); LOBO: length of bowel open.

**Table 6 nutrients-12-00264-t006:** Comparison of postoperative complications of surgical GC (gynecologic cancer) between intervention (CHO-P) and control groups (CO).

Postoperative Complication	CHO-P (*n* = 62)	CO (*n* = 56)	*p-*Value
	*n* (%)	*n* (%)	
**Postoperative nausea**			
Yes	16 (27)	44 (73)	0.000 ***
No	46 (79)	12 (21)	
**Postoperative vomiting**			
Yes	11 (24)	34 (76)	0.000 ***
No	51 (70)	22 (30)	
**Pneumonia**			
Yes	0 (0)	1 (100)	
No	62 (53)	55 (47)	
**Postoperative Ileus**			
Yes	0 (0)	1 (100)	
No	62 (53)	55 (47)	
**Infection**			
Yes	1 (17)	5 (83)	
No	61 (54)	51 (46)	
**Readmission within 30 days post discharged**			
Yes			
No	4 (6)	9 (16)	0.031 *
	58 (94)	47 (84)	

Chi-square test. * *p* < 0.05; *** *p* < 0.001.

**Table 7 nutrients-12-00264-t007:** Comparison of postoperative nutritional, biochemical, and functional outcomes of surgical GC patients between intervention (CHO-P) and control groups (CO).

Characteristics	Changes Post-Operation		*p*-Value
	CHO-P (*n* = 62) (mean ± SD)	CO (*n* = 56) (mean ± SD)	
**Body composition**			
Weight (kg)	−0.3 ± 2.3	−2.1 ± 2.3	<0.001 ***
BMI (kg/m^2^)	−0.1 ± 3.5	−1.2 ± 2.8	0.034 *
Muscle mass (kg)	0.4 ± 1.7	−0.7 ± 2.6	0.007 **
FM (kg)	−0.8 ± 2.2	−1.8 ± 2.0	0.010 **
FFM (kg)	−0.3 ± 19.2	−0.6 ± 2.6	0.035 *
MUAC (cm)	−1.5 ± 1.2	−1.9 ± 1.3	0.698
**Biochemical profile**			
Albumin (g/L)	−7.6 ± 4.9	−9.3 ± 6.1	0.110
C-reactive protein (mg/L)	5.0 ± 33.7	22.3 ± 59.8	0.050 *
CAR	0.3 ± 1.2	1.1 ± 2.6	0.030 *
Hemoglobin (g/L)	−1.0 ± 1.3	−0.3 ± 1.9	0.018 *
**Functional status**			
Handgrip strength (kg)	0.6 ± 4.3	−1.9 ± 4.7	0.004 *

Independent *t*-test; * *p* < 0.05; ** *p* < 0.01; *** *p* < 0.001. Abbreviations: BMI: body mass index; FM: fat mass; FFM: fat-free mass; MUAC: mid-upper arm circumference.

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
