# Peer review of "Impact of Enhanced Recovery after Surgery with Preoperative Whey Protein-Infused Carbohydrate Loading and Postoperative Early Oral Feeding among Surgical Gynecologic Cancer Patients: An Open-Labelled Randomized Controlled Trial"

_nutrients, 2020, doi:10.3390/nu12010264_

Round 1

Reviewer 1 Report

I applaud the authors' efforts to conduct this RCT within ERAS Gyn Onc where there are very few.  I believe the manuscript would be strengthened if the following issues are addressed:

the term Fast Track is an antiquated term, this should be substituted with ERAS throughout the text a lot of the language used is unconventional and awkward, I would suggest an English language medical editor review the text your FTR (ERAS) protocol has very few components compared to the accepted standard ERAS protocol for Gyn Onc (ref 17); why did you not use a more comprehensive protocol? you need to report the compliance for each of the components in your FTR (ERAS) protocol - if this cannot be done, explain why and how this impacts your results (see minimum requirements for reporting ERAS related research in: Elias K et al, The Reporting on ERAS Compliance, Outcomes, and Elements Research (RECOvER) Checklist: A Joint Statement by the ERAS® and ERAS® USA Societies. World J Surg. 2019 Jan;43(1):1-8.) how do you know your results are due to preop whey protein infused CHO loading/POEOF and not some other component of your FTR (ERAS) protocol?

Reviewer 2 Report

Thank you for giving me the opportunity to review the manuscript nutrients-666578, with the title "Impact of Preoperative Whey Protein Infused Carbohydrate Loading and Post-Operative Early Oral Feeding among Surgical Gynaecologic Cancer Patients under Fast Track Recovery Surgery: An Open Labelled Randomized Controlled Trial". This is a prospective, open labelled randomized controlled trial (RCT) in Malaysia. This manuscript concerns an important topic to evaluate the protocol of sole carbohydrate (CHO) loading and post-operative early oral feeding in surgical gynaecologic cancer patients. The study design is reasonable to practice and the research outcomes are clear.

I have some suggestions to provide the author to let the manuscript more easily to read by the audience.  

Table 6: please correct the misalignment Discussion: Please re-arranged the order in the part of discussion as following: the main findings, discussion of primary outcomes, secondary outcomes and other outcomes. Then summarized the strength and limitation of current study. Discussion of aspiration pneumonia could be listed as the secondary outcome.

Round 2

Reviewer 1 Report

The authors have done a good job of addressing the issues.  The only final issue that needs to be addressed:

- on lines 49-51 the text indicates: "Kehlet and Wilmore developed a 
multimodal perioperative protocol, named Enhanced Recovery after Surgery (ERAS) [5]."  This in fact is not correct, Kehlet developed Fast Track Surgery which can still be left in the text at this one location.  You should indicate right afterwards that "these concepts later led to the development of ERAS by   Ljungqvist and colleagues [ref Enhanced Recovery After Surgery: A Review.  Ljungqvist O, Scott M, Fearon KC. JAMA Surg. 2017 Mar 1;152(3):292-298.]

Author Response

Thank you for your insightful comments. We have revised  the statements in line 49-51 as per recommended. The track changes highlighted in blue.